# The effect of calcium supplementation in people under 35 years old: A systematic review and meta-analysis of randomized controlled trials

**Yupeng Liu[1†], Siyu Le[2†], Yi Liu[2], Huinan Jiang[2], Binye Ruan[2], Yufeng Huang[2], Xuemei Ao[2], Xudong Shi[2], Xiaoyi Fu[2]\*, Shuran Wang[2]\***

[1]Department of Epidemiology and Biostatistics, School of Public Health and Management, Wenzhou Medical University, Wenzhou, China; [2]Department of Nutrition and Food Hygiene, School of Public Health and Management, Wenzhou Medical University, Wenzhou, China

**\*For correspondence:**
fuxiaoyi@wmu.edu.cn (XF);
shuranwang@wmu.edu.cn (SW)

[†]These authors contributed equally to this work

**Competing interest:** The authors declare that no competing interests exist.

## Abstract

**Background:** The effect of calcium supplementation on bone mineral accretion in people under 35 years old is inconclusive. To comprehensively summarize the evidence for the effect of calcium supplementation on bone mineral accretion in young populations (≤35 years).

**Methods:** This is a systematic review and meta-analysis. The Pubmed, Embase, ProQuest, CENTRAL, WHO Global Index Medicus, Clinical Trials.gov, WHO ICTRP, China National Knowledge Infrastructure (CNKI), and Wanfang Data databases were systematically searched from database inception to April 25, 2021. Randomized clinical trials assessing the effects of calcium supplementation on bone mineral density (BMD) or bone mineral content (BMC) in people under 35 years old.

**Results:** This systematic review and meta-analysis identified 43 studies involving 7,382 subjects. Moderate certainty of evidence showed that calcium supplementation was associated with the accretion of BMD and BMC, especially on femoral neck (standardized mean difference [SMD] 0.627, 95% confidence interval [CI] 0.338–0.915; SMD 0.364, 95% CI 0.134–0.595; respectively) and total body (SMD 0.330, 95% CI 0.163–0.496; SMD 0.149, 95% CI 0.006–0.291), also with a slight improvement effect on lumbar spine BMC (SMD 0.163, 95% CI 0.008–0.317), no effects on total hip BMD and BMC and lumbar spine BMD were observed. Very interestingly, subgroup analyses suggested that the improvement of bone at femoral neck was more pronounced in the peripeak bone mass (PBM) population (20–35 years) than the pre-PBM population (<20 years).

**Conclusions:** Our findings provided novel insights and evidence in calcium supplementation, which showed that calcium supplementation significantly improves bone mass, implying that preventive calcium supplementation before or around achieving PBM may be a shift in the window of intervention for osteoporosis.

**Funding:** This work was supported by Wenzhou Medical University grant [89219029].

## Editor's evaluation

This manuscript is of interest to researchers and practitioners who are researching or treating osteoporosis. The effect of calcium supplementation on bone mineral density improvement, which was not shown in the elderly or children was shown in subjects younger than 35 years of age near PBM. It provides an important conclusion that calcium supplementation should be seriously considered at that age to prevent osteoporosis.

**eLife digest** Osteoporosis and bone fractures are common problems among older people, particularly older women. These conditions cause disability and reduce quality of life. Progressive loss of bone mineral density is usually the culprit. So far, strategies to prevent bone weakening with age have produced disappointing results. For example, taking calcium supplements in later life only slightly reduces the risk of osteoporosis or fracture. New approaches are needed.

Bone mass increases gradually early in life and peaks and plateaus around 20-35 years of age. After that period, bone mass slowly declines. Some scientists suspect that increasing calcium intake during this period of peak bone mass may reduce osteoporosis or fracture risk later in life.

A meta-analysis by Liu, Le et al. shows that boosting calcium intake in young adulthood strengthens bones. The researchers analyzed data from 43 randomized controlled trials that enrolled 7,382 participants. About half the studies looked at the effects of taking calcium supplements and the other half analyzed the effects of a high calcium diet. Boosting calcium intake in people younger than age 35 improved bone mineral density throughout the body. It also increased bone mineral density at the femoral neck, where most hip fractures occur. Calcium supplementation produced larger effects in individuals between the ages of 20 and 35 than in people younger than 20.

Both high calcium diets and calcium supplements with doses less than 1000 mg/d boosted bone strength. Higher dose calcium supplements did not provide any extra benefits. The analysis suggests people should pay more attention to bone health during early adulthood. Large randomized clinical trials are needed to confirm the long-term benefits of boosting calcium intake during early adulthood. But if the results are validated, taking calcium supplements, or eating more calcium-rich foods between the ages of 20 and 35 may help individuals build healthier bones and prevent fractures and osteoporosis later in life.

## Introduction

Osteoporosis is an imperative public health problem, particularly in elderly women (*Anonymous, 1993*; *Jones et al., 1994*; *Si et al., 2015*). Low bone mass and a fast rate of bone loss at menopause are equal risk factors for future fracture (*Riis et al., 1996*). A low bone mineral content (BMC) or bone mineral density (BMD) in an elderly person implies a suboptimal bone mass in young adulthood – related to peak bone mass (PBM), greater bone loss in later life, or both. A number of studies have concluded that increasing calcium intake in older people is unlikely to translate into clinically meaningful reductions in fractures or produce progressive increases in bone mass (*Tai et al., 2015*; *Zhao et al., 2017*; *Bolland et al., 2015*; *Hu et al., 2019*). It seems that calcium supplementation is meaningless in the elderly. On the other hand, intervention before the achievement of PBM to maximize PBM might have a significant influence on bone health and further prevent osteoporosis later in life. Several clinical trials have shown positive effects of calcium supplementation on BMD or BMC in children (*Lloyd et al., 1993*; *Khadilkar et al., 2012*). However, several clinical trials have concluded that calcium supplementation may not be associated with calculated bone mass or strength (*Lu et al., 2019*; *Vogel et al., 2017*). Narrative reviews have also concluded that calcium supplementation may have small nonprogressive effects on BMD or BMC (*Winzenberg et al., 2006*; *Huncharek et al., 2008*). To summarize the studies above, there have been considerable debates about whether calcium supplementation has effects on bone health among young people.

Very recently, a study using cross-sectional data from NHANES 2005–2014 concluded that the age at attainment of peak femoral neck BMD, total hip BMD, and lumbar spine BMD was 20–24 years old in males and 19–20 years old in females (*Xue et al., 2020*). Additionally, a plateau is achieved in PBM at approximately 30 years old (*Baxter-Jones et al., 2011*). Based on the literature above, we decided to limit the threshold to 35 years old in a conservative manner. Since the results of studies in young people are controversial, we carried out a comprehensive meta-analysis to determine the effectiveness of calcium supplementation for improving BMD or BMC in young people before the age of 35. We also aimed to determine whether any effect would vary by sex, baseline calcium intake, ethnicity, age or sources, duration, and doses of calcium supplementation.

## Methods

This meta-analysis was reported according to Preferred Reporting Items for Systematic Reviews and Meta-analyses guidelines (*Liberati et al., 2009*). The protocol for this meta-analysis is available in PROSPERO (CRD42021251275).

### Literature search

We applied search strategies to the following electronic bibliographic databases without language restrictions: PubMed, EMBASE, ProQuest, CENTRAL (Cochrane Central Register of Controlled Trials), WHO Global Index Medicus, Clinical Trials.gov, WHO ICTRP, China National Knowledge Infrastructure (CNKI), and Wanfang Data in April 2021 and updated the search in July 2022 for eligible studies addressing the effect of calcium or calcium supplementation, milk or dairy products with BMD or BMC as endpoints. Detailed search strategies are provided in *Supplementary file 1*. Only randomized controlled trials (RCTs) were included in this study. We also hand-searched conference abstract books. The reference sections and citation lists of the retrieved literature, including original research articles, reviews, editorials, and letters, were reviewed for potentially relevant articles.

### Inclusion criteria

We selected trials based on the following criteria: (1) RCTs comparing calcium or calcium plus vitamin D supplements with a placebo or no treatment; (2) trials involving participants aged under 35 years at baseline; (3) trials providing BMD (g/cm$^2$) or BMC (g) data measured by dual energy X-ray absorptiometry as estimates of bone mass. Exclusion criteria: (1) observational studies, such as cohorts, case–control studies, or cross-sectional studies; (2) participants aged over 35 years; (3) trials of participants who were pregnant or in the lactation period; (4) trials without a placebo or control group; (5) trials supplied with only vitamin D; (6) trials that had essential data missing. Two authors (YPL and SYL) independently screened titles and abstracts, and then full texts of relevant articles according to the inclusion and exclusion criteria. By thoroughly reading full texts, the reasons for excluded trials are provided in *Supplementary file 2*.

### Risk-of-bias assessments

The quality of the included RCTs was assessed independently by two reviewers (SYL and HNJ) based on the Revised Cochrane Risk-of-Bias Tool for Randomized Trials (RoB 2 tool, version August 22, 2019) (*Jpt, 2021*) and each item was graded as low risk, high risk, and some concerns. The five domains included the randomization process, deviations from intended interventions, missing outcome data, measurement of the outcome, and selection of the reported result. A general risk conclusion can be drawn from the risk assessment of the above five aspects. We defined the included trials as low, high, and moderate quality based on the overall bias, which is consistent with the RoB 2 tool algorithm. Disagreements were resolved by consensus.

### Data extraction and synthesis

Two researchers (YPL and SYL) independently used a structured data sheet to extract the following information from each study: authors, publication year, participant characteristics, doses of the supplements, baseline dietary calcium intake, duration of trials, and follow-up. The absolute changes in BMD or BMC at the lumbar spine, femoral neck, total hip, and total body were the primary outcomes we extracted. We categorized the studies into two groups by duration: <18 months and ≥18 months. For studies that presented the percentage change rather than absolute data, we calculated the absolute change value using baseline data, and the standard deviation and percentage change from baseline were consistent with the approach described in the Cochrane Handbook (*Jpt, 2021*). If there was missing information, we contacted the corresponding author and obtained the data. (If no reply was received for over 3 months, we would exclude the article.)

### Statistical analysis

The association of calcium with or without vitamin D supplements with BMD and BMC was assessed. We pooled the data (study level) from each study using random-effects models in a conservative manner. The standardized mean difference (SMD) and corresponding 95% confidence intervals (CIs) were reported. We performed predesigned subgroup analyses based on the following aspects: sex (female

vs. male) and age at baseline (<20 vs.≥20 years, representing the prepeak and peripeak subgroups, respectively; all analyzed trials were divided into two groups by the age of achieving PBM [determined as 20 years old]), regions (Asian and Western), sources of calcium supplementation (dietary vs. calcium supplements), and bias risk of each individual trial. We further conducted some post hoc subgroup analyses according to the level of calcium intake at baseline (<714 vs.≥714 mg/day, based on the median value), the calcium supplementation dose (<1000 vs. ≥1000 mg/day, based on the median value) and vitamin D supplementation (with or without vitamin D). To assess how long the beneficial effect would be maintained, we performed post hoc subgroup analyses according to the duration, taking into account different calcium supplementation periods and different follow-up periods across the trials. Sensitivity analyses included evaluations using fixed-effect models and excluding low-quality trials. In these aforementioned subgroup analyses, if the number of eligible studies in subgroups was less than three, we conducted a sensitivity analysis by excluding the subgroup with fewer than three studies. An effect size of ≥0.20 and<0.50 was considered small, ≥0.50 and <0.80 was considered medium, and ≥0.80 was considered large using Cohen's criteria (*Cohen, 1992*).

We assessed heterogeneity between studies using the $I^2$ statistic. We performed meta-regression for sample size, age, sex, and supplementation differences to explain the heterogeneity between studies. We performed cumulative meta-analyses based on the sample size to compare with the primary outcomes. We assessed publication bias by examining funnel plots when the number of trials was 10 or more and used Begg's rank correlation and Egger's linear regression tests (*Egger et al., 1997*). Furthermore, we robustly adjusted for the summarized results by applying Duval and Tweedie's trim and fill method (*Duval and Tweedie, 2000*). Data extraction and integration were done on Microsoft Office Excel (version 2011). Meta-analysis, subgroup analysis and sensitivity analysis were all performed by Comprehensive Meta Analysis (version 3.3.070, Biostat, Englewood, NJ). All tests were two tailed, and p < 0.05 was considered statistically significant. Two reviewers (SYL and YL) independently applied the Grading of Recommendations Assessment, Development and Evaluation (GRADE) system to assess the overall quality of evidence. The quality of evidence for each outcome was classified as either high, moderate, low, or very low based on the evaluation for study design, bias risk, inconsistency, indirectness, imprecision, publication bias, and confounding bias. GRADE pro version 3.6 was used to grade the overall quality of evidence and prepare the summary-of-findings table. Every decision to downgrade or upgrade the studies was labeled using footnotes. Any disagreements were resolved by consensus.

## Role of the funding source

The funders had no role in study design, data collection, data analysis, data interpretation, or writing of the report.

## Results

### Study characteristics

Of the 5518 references screened, we identified 43 eligible RCTs (*Figure 1*) involving 7382 subjects (*Lloyd et al., 1993*; *Khadilkar et al., 2012*; *Lu et al., 2019*; *Vogel et al., 2017*; *Bonjour et al., 1997*; *Cadogan et al., 1997*; *Cameron et al., 2004*; *Cheng et al., 2005*; *Chevalley et al., 2005a*; *Du et al., 2004*; *Gibbons et al., 2004*; *Lau et al., 2004*; *Lee et al., 1995*; *Lee et al., 1994*; *Lloyd et al., 1996*; *Matkovic et al., 2005*; *Moyer-Mileur et al., 2003*; *Prentice et al., 2005*; *Rozen et al., 2003*; *Specker and Binkley, 2003*; *Stear et al., 2003*; *Courteix et al., 2005*; *Iuliano-Burns et al., 2003*; *Johnston et al., 1992*; *Mølgaard et al., 2004*; *Nowson et al., 1997*; *Ho et al., 2005*; *Ma et al., 2014*; *Zhang et al., 2014*; *Ward et al., 2014*; *Arab Ameri et al., 2012*; *Ekbote et al., 2011*; *Hemayattalab, 2010*; *Islam et al., 2010*; *Yin et al., 2010*; *Lambert et al., 2008*; *Zhu et al., 2008*; *Ward et al., 2007*; *Bass et al., 2007*; *Barger-Lux et al., 2005*; *Chevalley et al., 2005b*; *Winters-Stone and Snow, 2004*; *Volek et al., 2003*). *Table 1* shows the baseline characteristics of the included studies. Of the 43 RCTs, 20 used dietary sources of calcium (*Lu et al., 2019*; *Vogel et al., 2017*; *Bonjour et al., 1997*; *Cadogan et al., 1997*; *Cheng et al., 2005*; *Du et al., 2004*; *Gibbons et al., 2004*; *Lau et al., 2004*; *Iuliano-Burns et al., 2003*; *Nowson et al., 1997*; *Ho et al., 2005*; *Ma et al., 2014*; *Zhang et al., 2014*; *Arab Ameri et al., 2012*; *Ekbote et al., 2011*; *Lambert et al., 2008*; *Zhu et al., 2008*; *Bass et al., 2007*; *Volek et al., 2003*) and 23 used calcium supplements (including calcium, calcium citrate

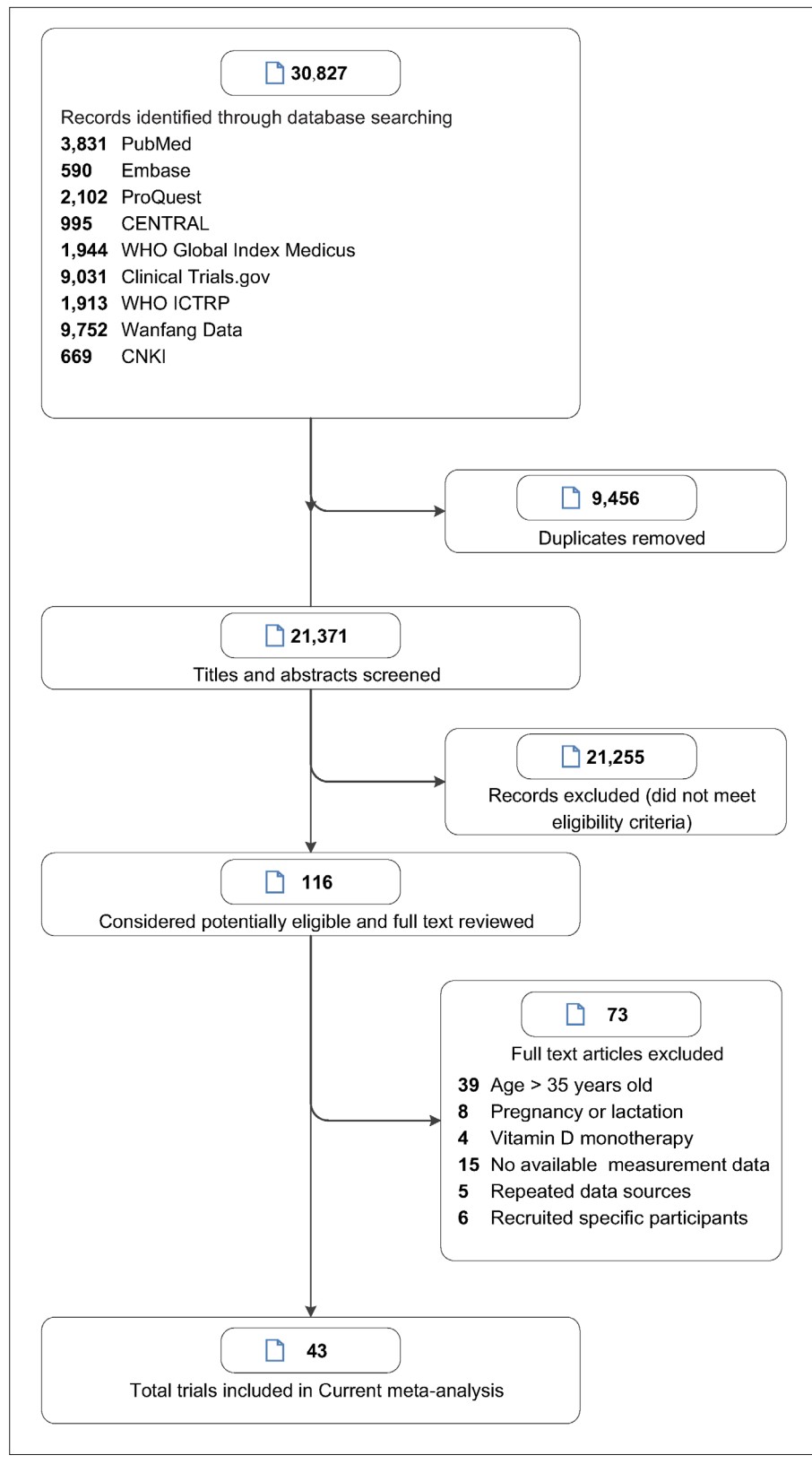

**Figure 1.** Study selection.

**Table 1.** Characteristics of included studies.

| Study | Supplement and Ca dose (mg/day) | Duration of supplement/ follow-up (years) | No. of subjects | Ethnicity | Female (%) | Mean (SD or range) age (years) | Mean baseline Cacium intake (mg/day) | Site measured |
|---|---|---|---|---|---|---|---|---|
| *Bonjour et al., 1997* | Milk extract, 850 | 1/2 | 144 | White | 100 | 7.94 ± 0.1 | 912 ± 42 | Radius, hip, LS |
| *Cadogan et al., 1997* | Whole or reduced fat milk, 1125 | 1.5/1.5 | 82 | White | 100 | 12.2 ± 0.3 | 746 | TB |
| *Cameron et al., 2004* | CaCO₃, 1200 | 2/2 | 128 | White | 100 | 10.3 ± 0.2 | 715 | LS, forearm, hip, TB |
| *Cheng et al., 2005* | CaCO₃ or dairy products, 1000 | 2/2 | 181 | White | 100 | 11 (10–12) | <900 | LS, FN, TB |
| *Chevalley et al., 2005b* | CaPO₄, 850 | 1/2 | 235 | White | 0 | 7.4 ± 0.4 | 750 | Radius, hip, LS |
| *Du et al., 2004* | Milk, 245 | 2/2 | 757 | Chinese | 100 | 11 (10–12) | 418 | Forearm, TB |
| *Gibbons et al., 2004* | Dairy drink, 1200 | 1.5/2.5 | 154 | White | 51 | 9 (8–10) | 934 | TB, hip, LS |
| *Lau et al., 2004* | Milk powder, 650 or 1300 | 1.5/1.5 | 344 | Chinese | 45 | 8 (9–10) | 463 | Hip, LS, TB |
| *Lee et al., 1995* | CaCO₃, 300 | 1.5/1.5 | 109 | Chinese | 42 | Age 7 | 567 | Radius, LS, FN |
| *Lee et al., 1994* | CaCO₃, 300 | 1.5/1.5 | 162 | Chinese | 46 | Age 7 | 280 | Radius |
| *Lloyd et al., 1993* | CaCM, 500 | 2/2 | 94 | White | 100 | 11.9 ± 0.5 | 960 | LS, TB |
| *Lloyd et al., 1996* | CaCM, 500 | 2/2 | 112 | White | 100 | 11.9 ± 0.5 | 983 | LS, TB |
| *Matkovic et al., 2005* | CaCM, 1000 | 7/7 | 354 | White | 100 | Age 11 | 830 | Radius, TB |
| *Moyer-Mileur et al., 2003* | CaCO₃, 800 | 1/1 | 71 | White | 100 | Age 12 | 900 | TB, trabecular |
| *Prentice et al., 2005* | CaCO₃, 1000 | 1/1 | 143 | White | 0 | 16.8 (16–18) | 1190 | TB, LS, hip, forearm |

*Table 1 continued on next page*

*Table 1 continued*

| Study | Supplement and Ca dose (mg/day) | Duration of supplement/follow-up (years) | No. of subjects | Ethnicity | Female (%) | Mean (SD or range) age (years) | Mean baseline Cacium intake (mg/day) | Site measured |
|---|---|---|---|---|---|---|---|---|
| **Rozen et al., 2003** | Elemental calcium, 1000 | 1/1 | 112 | 76% Jewish girls, 24% Arab | 100 | 14 ± 0.5 | 580 | TB, LS, FN |
| **Specker and Binkley, 2003** | CaCO$_3$, 1000 | 1/1 | 178 | White | 47 | 4 (3–5) | 900 | TB, arm, leg |
| **Stear et al., 2003** | CaCO$_3$, 1000 | 1.3/1.3 | 144 | White | 100 | 17.3 ± 0.3 | 938 ± 411 | TB, LS, hip, forearm |
| **Courteix et al., 2005** | CaPO$_4$, 800 | 1/1 | 113 | White | 100 | 10 (8–13) | 980 | TB, LS, hip, radius |
| **Iuliano-Burns et al., 2003** | Food products fortified by milk minerals, 400 | 0.7/0.7 | 75 | 85% White, 15% Asian | 100 | 8.8 ± 0.1 | 673 | TB, LS, leg, arm |
| **Johnston et al., 1992** | CaCM, 1000 | 3/3 | 140 | White | 61 | 10 ± 2 | 908 | Radius, hip, LS |
| **Molgaard et al., 2004** | CaCO$_3$, 500 | 1/1 | 113 | White | 100 | 13.2 (12.6–13.7) | A: 1000–1307; B:<713 | TB |
| **Nowson et al., 1997** | CaCO$_3$/Ca-lactate gluconate, 1000 | 1.5/1.5 | 84 | White | 100 | 14 ± 2.6 | 750 | LS, hip, forearm, TB |
| **Ho et al., 2005** | Calcium-fortified soymilk supplementation, 600 | 1/1 | 210 | Chinese | 100 | 14.5 ± 0.39 | 510 | LS, hip |
| **Lu et al., 2019** | Milk powder, 300/600/900 | 1.5/1.5 | 232 | Chinese | 50 | 13 (12–15) | 370 | TB, LS, hip |
| **Vogel et al., 2017** | Dairy products, 900 | 1.5/1.5 | 240 | 61% Black, 35% White, 4% NS | 66 | 11.8 ± 1.5 | 700 | TB, hip |
| **Ma et al., 2014** | Milk powder, 300/600/900 | 1/1 | 220 | Chinese | 50 | 12.9 ± 0.3 | 700 | TB, LS, hip |
| **Zhang et al., 2014** | Milk powder or additional calcium, 300/600/900 | 2/2 | 220 | Chinese | 50 | 12.9 ± 0.3 | 700 | TB, LS, hip |

*Table 1 continued on next page*

Table 1 continued

| Study | Supplement and Ca dose (mg/day) | Duration of supplement/ follow-up (years) | No. of subjects | Ethnicity | Female (%) | Mean (SD or range) age (years) | Mean baseline Calcium intake (mg/day) | Site measured |
|---|---|---|---|---|---|---|---|---|
| *Ward et al., 2014* | CaCO₃ 1000 | 1/12 | 80 | Black | 0 | 12.5 ± 0.1 | 338 | LS, hip |
| *Khadilkar et al., 2012* | CaCO₃ 500 | 1/1 | 210 | Indian | 100 | 9.9 ± 1.0 | 250 | TB |
| *Arab Ameri et al., 2012* | Milk, 250 | 0.75/0.75 | 54 | White | 0 | 10.3 ± 2.2 | 570 | FN |
| *Ekbote et al., 2011* | Calcium fortified laddoo, 405 mg | 1/1 | 60 | Indian | 50 | 2.7 ± 0.52 | 188 | TB |
| *Hemayattalab, 2010* | Milk, 230 | 0.5/0.5 | 40 | White | 0 | 8.6 ± 1.1 | 480 | FN |
| *Islam et al., 2010* | Ca-lactate, 600 | 1/1 | 200 | White | 100 | 22.9 ± 3.9 | <500 | LS, hip |
| *Yin et al., 2010* | Calcium, 85/230/500 | 2/2 | 257 | Chinese | 47 | 13.5 ± 0.5 | 300 | TB, LS |
| *Lambert et al., 2008* | Calcium-fortified fruit drink, 792 | 1.5/3.5 | 89 | White | 100 | 11.41 ± 0.54 | 636 | TB, LS, hip |
| *Zhu et al., 2008* | Milk, 650 | 2/2 | 757 | Chinese | 100 | 10.1 ± 0.4 | 436 | TB |
| *Ward et al., 2007* | Elemental calcium, 500 | 1/1 | 75 | White | 60 | 9.8 ± 1.6 | 850 | TB, LS |
| *Bass et al., 2007* | Ca-fortified foods using milk minerals, 392 ± 29 | 0.7/0.7 | 88 | White | 0 | 9.0 ± 0.3 | 900 | TB, LS |
| *Barger-Lux et al., 2005* | CaCO₃ 500 | 3/3 | 121 | White | 100 | 23.1 ± 2.7 | 605 | TB, LS, hip |
| *Chevalley et al., 2005b* | Milk calcium-phosphate salt extract, 850 | 1/8 | 149 | White | 100 | 7.9 ± 0.5 | 900 | Radius, hip, LS |
| *Winters-Stone and Snow, 2004* | CaCO₃ 1000 | 1/1 | 23 | White | 100 | 23.7 ± 4.7 | 1100 | Hip, LS, femoral mid-shaft |
| *Volek et al., 2003* | Milk, 1723 ± 274 | 0.25/0.25 | 28 | White | 0 | 13–17 | 1000 | TB |

CaCO₃ = calcium carbonate; Ca = calcium; CaCM = calcium citrate malate; CaPO₄ = calcium phosphate; LS = lumbar spine; TB = total body; FN = femoral neck; NS = not stated.

**Figure 2. Effect of calcium supplementation on bone mineral density (BMD) in each sites**

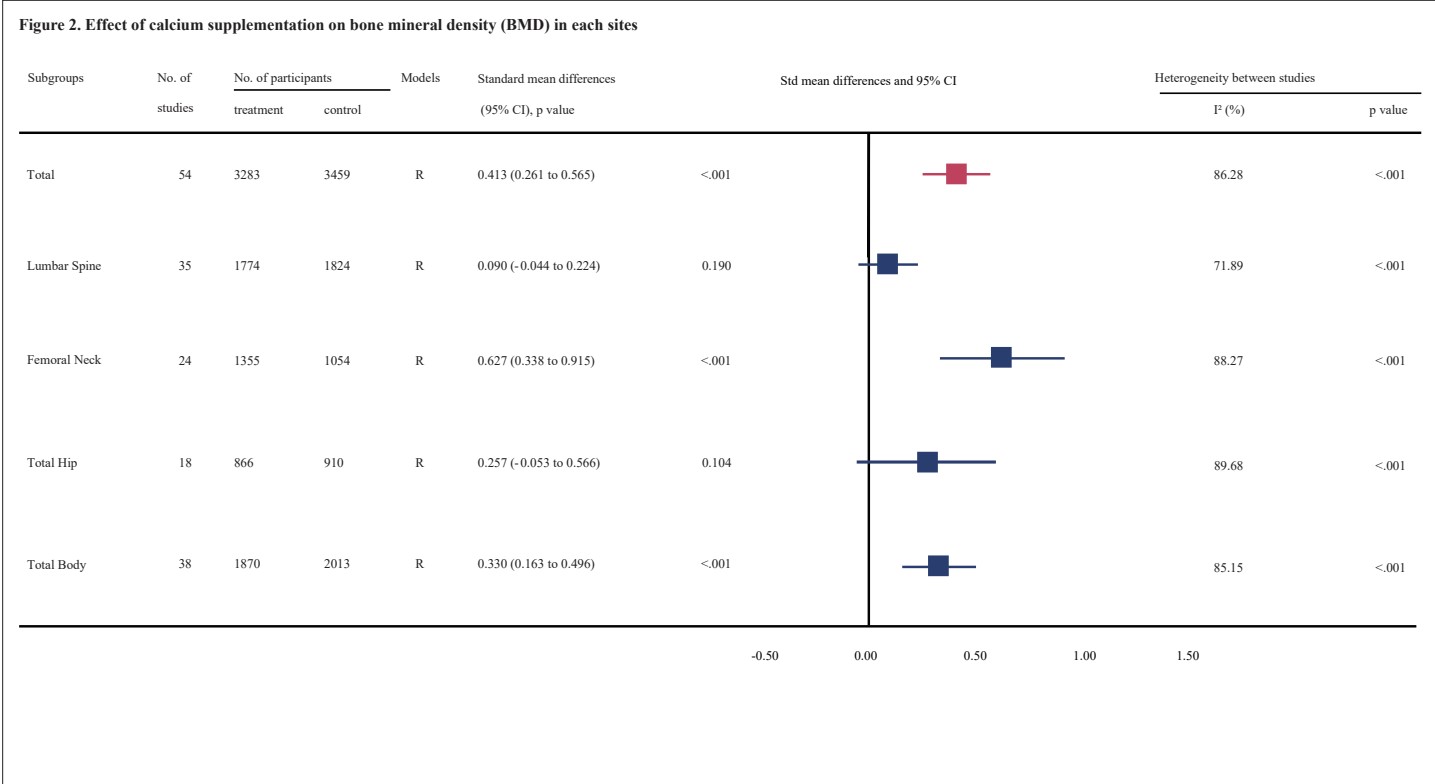

| Subgroups | No. of studies | No. of participants | | Models | Standard mean differences (95% CI), p value | | Std mean differences and 95% CI | Heterogeneity between studies | |
|---|---|---|---|---|---|---|---|---|---|
| | | treatment | control | | | | | I² (%) | p value |
| Total | 54 | 3283 | 3459 | R | 0.413 (0.261 to 0.565) | <.001 | | 86.28 | <.001 |
| Lumbar Spine | 35 | 1774 | 1824 | R | 0.090 (- 0.044 to 0.224) | 0.190 | | 71.89 | <.001 |
| Femoral Neck | 24 | 1355 | 1054 | R | 0.627 (0.338 to 0.915) | <.001 | | 88.27 | <.001 |
| Total Hip | 18 | 866 | 910 | R | 0.257 (- 0.053 to 0.566) | 0.104 | | 89.68 | <.001 |
| Total Body | 38 | 1870 | 2013 | R | 0.330 (0.163 to 0.496) | <.001 | | 85.15 | <.001 |

**Figure 2.** Effect of calcium supplmentation on bone mineral density (BMD) in each site.

The online version of this article includes the following source data for figure 2:

**Source data 1.** Forest plots for the association between calcium supplementation and the accretion of lumbar spine bone mineral density (LSBMD).

**Source data 2.** Forest plots for the association between calcium supplementation and the accretion of femoral neck bone mineral density (FNBMD).

**Source data 3.** Forest plots for the association between calcium supplementation and the accretion of total hip bone mineral density (THBMD).

**Source data 4.** Forest plots for the association between calcium supplementation and the accretion of total body bone mineral density (TBBMD).

malate, and calcium phosphate) (*Lloyd et al., 1993*; *Khadilkar et al., 2012*; *Cameron et al., 2004*; *Chevalley et al., 2005a*; *Lee et al., 1995*; *Lee et al., 1994*; *Lloyd et al., 1996*; *Matkovic et al., 2005*; *Moyer-Mileur et al., 2003*; *Prentice et al., 2005*; *Rozen et al., 2003*; *Specker and Binkley, 2003*; *Stear et al., 2003*; *Courteix et al., 2005*; *Johnston et al., 1992*; *Mølgaard et al., 2004*; *Ward et al., 2014*; *Hemayattalab, 2010*; *Islam et al., 2010*; *Yin et al., 2010*; *Ward et al., 2007*; *Barger-Lux et al., 2005*; *Winters-Stone and Snow, 2004*). The median baseline dietary calcium intake was 714 mg/day; the duration of calcium supplementation intervention did not exceed 2 years in most trials (38/43); and the dose of calcium intervention did not exceed 1000 mg/day in most trials (38/43). Of all the included trials, 23 trials were categorized as low risk of bias; 16, as moderate risk; and 4, as high risk (*Supplementary file 3*).

## Primary analyses

*Figure 2*, *Figure 2—source data 1*, *Figure 2—source data 2*, *Figure 2—source data 3*, *Figure 2— source data 4*, *Figure 3*, *Figure 3—source data 1*, *Figure 3—source data 2*, *Figure 3—source data*

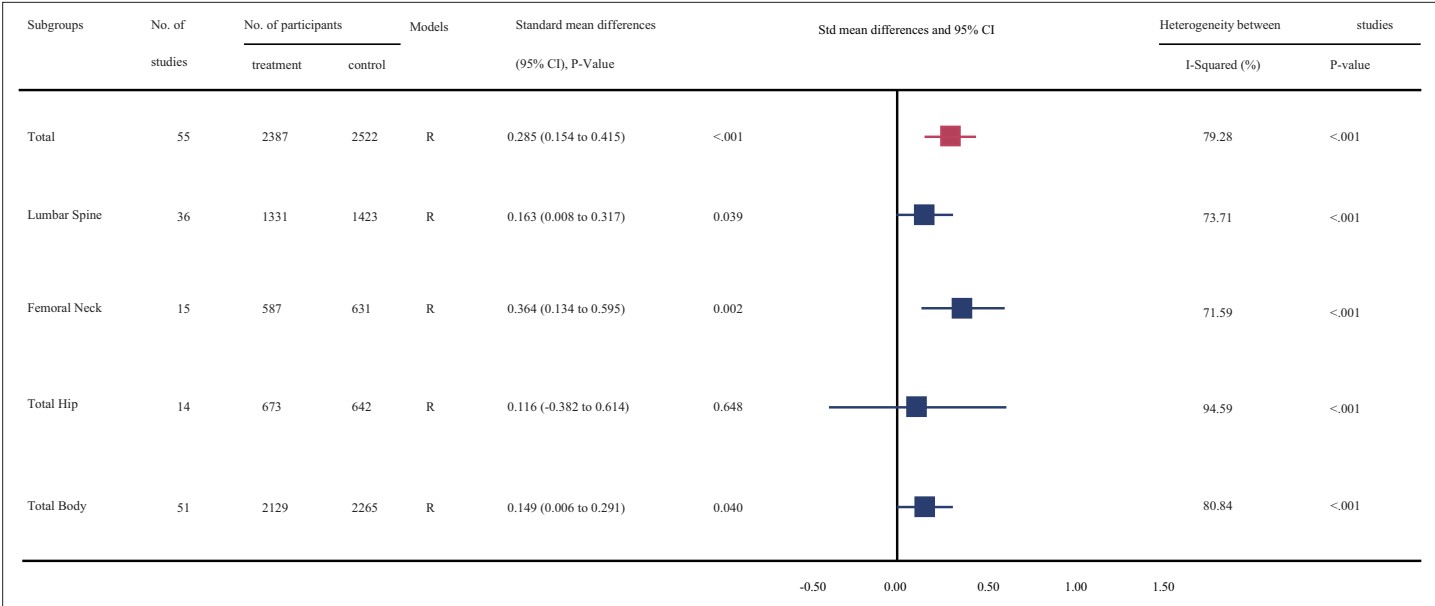

| Subgroups | No. of studies | No. of participants | | Models | Standard mean differences (95% CI), P-Value | | Std mean differences and 95% CI | Heterogeneity between studies | |
|---|---|---|---|---|---|---|---|---|---|
| | | treatment | control | | | | | I-Squared (%) | P-value |
| Total | 55 | 2387 | 2522 | R | 0.285 (0.154 to 0.415) | <.001 | | 79.28 | <.001 |
| Lumbar Spine | 36 | 1331 | 1423 | R | 0.163 (0.008 to 0.317) | 0.039 | | 73.71 | <.001 |
| Femoral Neck | 15 | 587 | 631 | R | 0.364 (0.134 to 0.595) | 0.002 | | 71.59 | <.001 |
| Total Hip | 14 | 673 | 642 | R | 0.116 (-0.382 to 0.614) | 0.648 | | 94.59 | <.001 |
| Total Body | 51 | 2129 | 2265 | R | 0.149 (0.006 to 0.291) | 0.040 | | 80.84 | <.001 |

**Figure 3.** Effect of calcium supplmentation on bone mineral content (BMC) in each site.

The online version of this article includes the following source data for figure 3:

**Source data 1.** Forest plots for the association between calcium supplementation and the accretion of lumbar spine bone mineral content (LSBMC).

**Source data 2.** Forest plots for the association between calcium supplementation and the accretion of femoral neck bone mineral content (FNBMC).

**Source data 3.** Forest plots for the association between calcium supplementation and the accretion of total hip bone mineral content (THBMC).

**Source data 4.** Forest plots for the association between calcium supplementation and the accretion of total body bone mineral content (TBBMC).

*3*, and *Figure 3—source data 4* show the summarized effect estimates. For total body, moderate evidence showed that calcium supplementation significantly improved BMD levels with an SMD of 0.330 (95% CI: 0.163–0.496, p < 0.001) and slightly improved BMC levels with an SMD of 0.149 (95% CI: 0.006–0.291, p < 0.001). At the femoral neck, we found a stronger and moderate protective effect on BMD (0.627, 95% CI: 0.338–0.915, p < 0.001) and a small improvement effect on BMC (0.364, 95% CI: 0.134–0.595, p = 0.002). Meanwhile, a slight but significant improvement in BMC was observed for the lumbar spine (0.163, 95% CI: 0.008–0.317, p = 0.039). However, calcium supplementation did not improve the BMD levels at the lumbar spine (0.090, 95% CI: −0.044 to 0.224, p = 0.190) or total hip (0.257, 95% CI: −0.053 to 0.566, p = 0.104) or the BMC level at the total hip (0.116, 95% CI: −0.382 to 0.614, p = 0.648).

## Subgroup analyses

*Tables 2 and 3* show the results of subgroup analyses. To explore whether the observed effect differed by the age of participants, we divided these participants into two subgroups: prepeak (<20 years) and peripeak (≥20–35 years), and the results were generally consistent with the findings from the primary analyses. Notably, the improvement effect on both BMD and BMC at the femoral neck (see *Figure 4*) tended to be stronger in the peripeak subjects than in the prepeak subjects (0.852, 95% CI: 0.257–1.446 vs. 0.600, 95% CI: 0.292–0.909 [for BMD] and 1.045, 95% CI: 0.701–1.39 vs. 0.249, 95% CI: 0.043–0.454 [for BMC], respectively).

Subgroup analyses by the duration of calcium supplementation showed that the improvement effects on both BMD and BMC of the femoral neck were stronger in the subgroup with <18 months than in the subgroup with ≥18 months. However, regarding total body BMD, the effect of calcium supplementation in the subgroup with ≥18 months duration was slightly greater than that in the other subgroup.

Regarding the sex of subjects, we found a stronger beneficial effect on femoral neck BMD and BMC in women-only trials (0.712, 95% CI: 0.149–1.275, p = 0.013; 0.742, 95% CI: 0.267–1.217, p =

**Table 2.** Subgroup analysis of bone mineral density (BMD) between calcium supplementation and control for each variable at lumbar spine, femoral neck, total hip, and total body.

| Variable | No. of datasets | No. of participants | BMD difference (95% CI), p value | Heterogeneity between studies | | p value* |
|---|---|---|---|---|---|---|
| | | | | I² (%) | p value | |
| Lumbar spine | | | | | | |
| Age | | | | | | |
| Prepeak | 31 | 3104 | 0.093 (−0.047 to 0.233), 0.192 | 71.54 | <0.001 | 0.866 |
| Peripeak | 4 | 344 | 0.078 (−0.471 to 0.627), 0.780 | 79.82 | 0.002 | |
| Duration | | | | | | |
| <18 months | 14 | 1420 | 0.066 (−0.069 to 0.202), 0.335 | 32.75 | 0.113 | 0.905 |
| ≥18 months | 21 | 2178 | 0.106 (−0.104 to 0.316), 0.322 | 80.31 | <0.001 | |
| Sex | | | | | | |
| Women-only trials | 13 | 1466 | 0.36 (0.067 to 0.653), 0.016 | 83.71 | <0.001 | 0.011 |
| Trials with men and women | 22 | 2181 | −0.057 (−0.162 to 0.048), 0.284 | 27.53 | 0.115 | |
| Regions | | | | | | |
| Asian | 18 | 1492 | −0.012 (−0.117 to −0.094), 0.829 | 12.70 | 0.302 | 0.177 |
| Western | 17 | 1956 | 0.222 (−0.03 to 0.473), 0.084 | 83.62 | <0.001 | |
| Baseline calcium intake, mg/day | | | | | | |
| <714 | 23 | 2014 | 0.062 (−0.109 to 0.234), 0.477 | 73.19 | <0.001 | 0.561 |
| ≥714 | 12 | 1434 | 0.145 (−0.080 to 0.370), 0.207 | 71.17 | <0.001 | |
| Calcium dose, mg/day | | | | | | |
| <1000 | 26 | 2172 | 0.103 (−0.062 to 0.269), 0.222 | 75.30 | <0.001 | 0.806 |
| ≥1000 | 9 | 1056 | 0.050 (−0.177 to 0.276), 0.667 | 59.22 | 0.012 | |
| Types of calcium supplement | | | | | | |

*Table 2 continued*

| Variable | No. of datasets | No. of participants | BMD difference (95% CI), p value | Heterogeneity between studies | | |
|---|---|---|---|---|---|---|
| | | | | $I^2$ (%) | p value | p value* |
| Dietary calcium | 18 | 1690 | 0.104 (−0.104 to 0.311), 0.328 | 77.83 | <0.001 | 0.870 |
| Calcium supplementation | 17 | 1758 | 0.075 (−0.099 to 0.249), 0.396 | 63.66 | <0.001 | |
| Supplementation with or without vitamin D | | | | | | |
| Without vitamin D | 22 | 2520 | 0.140 (−0.047 to 0.327), 0.143 | 78.59 | <0.001 | 0.468 |
| With vitamin D | 13 | 1078 | 0.008 (−0.160 to 0.176), 0.926 | 44.69 | 0.041 | |
| Femoral neck | | | | | | |
| Age | | | | | | |
| Prepeak | 21 | 1795 | 0.600 (0.292 to 0.909), <0.001 | 88.68 | <0.001 | 0.138 |
| Peripeak | 3 | 223 | 0.852 (0.257 to 1.446), 0.005 | 67.97 | 0.044 | |
| Duration | | | | | | |
| <18 months | 15 | 1457 | 0.824 (0.383 to 1.266), <0.001 | 91.06 | <0.001 | 0.578 |
| ≥18 months | 9 | 952 | 0.378 (0.047 to 0.709), 0.025 | 79.12 | <0.001 | |
| Sex | | | | | | |
| Women-only trials | 8 | 840 | 0.712 (0.149 to 1.275), 0.013 | 90.89 | <0.001 | 0.963 |
| Trials with men and women | 16 | 1262 | 0.560 (0.233 to 0.879), 0.001 | 85.41 | <0.001 | |
| Regions | | | | | | |
| Asian | 10 | 793 | 0.091 (−0.047 to 0.230), 0.197 | 0.00 | 0.441 | 0.115 |
| Western | 14 | 1309 | 1.078 (0.603 to 1.552), <0.001 | 91.53 | <0.001 | |
| Baseline calcium intake, mg/day | | | | | | |
| <714 | 17 | 1159 | 0.581 (0.266 to 0.896), <0.001 | 84.10 | <0.001 | 0.57 |
| ≥714 | 7 | 903 | 0.680 (0.036 to 1.323), 0.038 | 93.43 | <0.001 | |

*Table 2 continued on next page*

*Table 2 continued*

| Variable | No. of datasets | No. of participants | BMD difference (95% CI), p value | Heterogeneity between studies | | p value* |
|---|---|---|---|---|---|---|
| | | | | $I^2$ (%) | p value | |
| Calcium dose, mg/day | | | | | | |
| <1000 | 18 | 1371 | 0.717 (0.349 to 1.085), <0.001 | 89.52 | <0.001 | 0.488 |
| ≥1000 | 6 | 731 | 0.421 (−0.055 to 0.897), 0.083 | 85.12 | <0.001 | |
| Types of calcium supplement | | | | | | |
| Dietary calcium | 15 | 1071 | 0.728 (0.311 to 1.144), 0.001 | 89.73 | <0.001 | 0.635 |
| Calcium supplementation | 9 | 1031 | 0.510 (0.101 to 0.919), 0.014 | 86.60 | <0.001 | |
| Supplementation with or without vitamin D | | | | | | |
| Without vitamin D | 10 | 1331 | 0.477 (0.045 to 0.910), 0.031 | 91.44 | <0.001 | 0.119 |
| With vitamin D | 14 | 794 | 0.758 (0.350 to 1.166), <0.001 | 85.38 | <0.001 | |
| Total hip | | | | | | |
| Age | | | | | | |
| Prepeak | 16 | 1539 | 0.336 (0.031 to 0.642), 0.031 | 88.43 | <0.001 | 0.119 |
| Peripeak | 2 | 144 | −0.465 (−1.409 to 0.479), 0.334 | 77.90 | 0.033 | |
| Duration | | | | | | |
| <18 months | 6 | 485 | 0.076 (−0.102 to 0.255), 0.402 | 0.00 | 0.963 | 0.935 |
| ≥18 months | 12 | 1291 | 0.351 (−0.102 to 0.805), 0.129 | 93.24 | <0.001 | |
| Sex | | | | | | |
| Women-only trials | 5 | 527 | 0.483 (−0.479 to 1.444), 0.325 | 95.75 | <0.001 | 0.932 |
| Trials with men and women | 13 | 1070 | 0.181 (−0.103 to 0.465), 0.211 | 83.03 | <0.001 | |
| Regions | | | | | | |

*Table 2 continued on next page*

*Table 2 continued*

| Variable | No. of datasets | No. of participants | BMD difference (95% CI), p value | Heterogeneity between studies | | |
|---|---|---|---|---|---|---|
| | | | | I² (%) | p value | p value* |
| Asian | 13 | 1126 | 0.096 (–0.127 to 0.319), 0.399 | 73.92 | <0.001 | 0.579 |
| Western | 5 | 471 | 0.690 (–0.429 to 1.81), 0.227 | 96.33 | <0.001 | |
| Baseline calcium intake, mg/day | | | | | | |
| <714 | 15 | 1336 | 0.179 (–0.148 to 0.507), 0.283 | 89.55 | <0.001 | 0.023 |
| ≥714 | 3 | 261 | 0.723 (0.245 to 1.201), 0.003 | 60.02 | 0.082 | |
| Calcium dose, mg/day | | | | | | |
| <1000 | 14 | 1092 | 0.189 (–0.179 to 0.557), 0.314 | 90.28 | <0.001 | 0.329 |
| ≥1000 | 4 | 505 | 0.513 (–0.024 to 1.05), 0.061 | 84.04 | <0.001 | |
| Types of calcium supplement | | | | | | |
| Dietary calcium | 15 | 1369 | 0.314 (–0.006 to 0.634), 0.054 | 88.89 | <0.001 | 0.421 |
| Calcium supplementation | 3 | 228 | –0.046 (–1.148 to 1.056), 0.935 | 92.84 | <0.001 | |
| Supplementation with or without vitamin D | | | | | | |
| Without vitamin D | 7 | 894 | 0.506 (–0.138 to 1.149), 0.123 | 94.78 | <0.001 | 0.546 |
| With vitamin D | 11 | 878 | 0.101 (–0.191 to 0.393), 0.498 | 78.22 | <0.001 | |
| Total body | | | | | | |
| Age | | | | | | |
| Prepeak | 38 | 3883 | 0.330 (0.163 to 0.496), <0.001 | 85.15 | <0.001 | · |
| Peripeak | · | | · | · | · | · |
| Duration | | | | | | |
| <18 months | 12 | 986 | 0.324 (0.035 to 0.614), 0.028 | 79.55 | <0.001 | 0.775 |
| ≥18 months | 26 | 2897 | 0.334 (0.129 to 0.539), 0.001 | 87.15 | <0.001 | |

*Table 2 continued*

| Variable | No. of datasets | No. of participants | BMD difference (95% CI), p value | Heterogeneity between studies | | p value* |
|---|---|---|---|---|---|---|
| | | | | I² (%) | p value | |
| Sex | | | | | | |
| Women-only trials | 18 | 2359 | 0.569 (0.328 to 0.810), <0.001 | 87.66 | <0.001 | 0.036 |
| Trials with men and women | 20 | 1558 | 0.104 (−0.089 to 0.296), 0.292 | 73.86 | <0.001 | |
| Ethnicity | | | | | | |
| Asian | 23 | 2008 | 0.274 (0.062 to 0.486), 0.011 | 85.67 | <0.001 | 0.544 |
| Western | 15 | 1469 | 0.422 (0.143 to 0.701), 0.003 | 85.28 | <0.001 | |
| Baseline calcium intake, mg/day | | | | | | |
| <714 | 26 | 2356 | 0.363 (0.127 to 0.599), 0.003 | 89.23 | <0.001 | 0.140 |
| ≥714 | 12 | 1215 | 0.265 (0.136 to 0.394), <0.001 | 22.28 | 0.225 | |
| Calcium dose, mg/day | | | | | | |
| <1000 | 27 | 2612 | 0.392 (0.161 to 0.624), 0.001 | 88.51 | <0.001 | 0.484 |
| ≥1000 | 11 | 1285 | 0.189 (0.073 to 0.306), 0.001 | 11.81 | 0.332 | |
| Types of calcium supplement | | | | | | |
| Dietary calcium | 24 | 2453 | 0.290 (0.054 to 0.526), 0.016 | 88.33 | <0.001 | 0.129 |
| Calcium supplementation | 14 | 1464 | 0.405 (0.195 to 0.615), <0.001 | 74.22 | <0.001 | |
| Supplementation with or without vitamin D | | | | | | |
| Without vitamin D | 22 | 2657 | 0.701 (0.327 to 1.076), <0.001 | 94.83 | <0.001 | 0.137 |
| With vitamin D | 15 | 1625 | 0.156 (−0.156 to 0.468), 0.327 | 88.94 | <0.001 | |

*p value for heterogeneity between subgroups.

**Table 3.** Subgroup analysis of bone mineral content (BMC) between calcium supplementation and control for each variable at lumbar spine, femoral neck, total hip, and total body.

| Variable | No. of datasets | No. of participants | BMD difference (95% CI), p value | Heterogeneity between studies | | p value* |
|---|---|---|---|---|---|---|
| | | | | $I^2$ (%) | p value | |
| **Lumbar spine** | | | | | | |
| Age | | | | | | |
| Prepeak | 33 | 2465 | 0.173 (0.006 to 0.341), 0.043 | 75.06 | <0.001 | 0.678 |
| Peripeak | 3 | 321 | 0.047 (−0.291 to 0.384), 0.786 | 47.68 | 0.148 | |
| Duration | | | | | | |
| <18 months | 21 | 1485 | 0.063 (−0.063 to 0.190), 0.328 | 25.21 | 0.143 | 0.487 |
| ≥18 months | 15 | 1296 | 0.293 (−0.015 to 0.602), 0.062 | 82.27 | <0.001 | |
| Sex | | | | | | |
| Women-only trials | 14 | 1220 | 0.327 (−0.017 to 0.672), 0.062 | 86.55 | <0.001 | 0.496 |
| Trials with men and women | 22 | 1566 | 0.076 (−0.054 to 0.207), 0.251 | 38.52 | 0.035 | |
| Regions | | | | | | |
| Asian | 15 | 1260 | 0.003 (−0.108 to 0.113), 0.962 | 0.00 | 0.704 | 0.112 |
| Western | 21 | 1199 | 0.319 (0.059 to 0.579), 0.016 | 82.06 | <0.001 | |
| Baseline calcium intake, mg/day | | | | | | |
| <714 | 24 | 2030 | 0.137 (−0.075 to 0.349), 0.206 | 81.04 | <0.001 | 0.104 |
| ≥714 | 12 | 756 | 0.206 (0.059 to 0.354), 0.006 | 0.00 | 0.472 | |
| Calcium dose, mg/day | | | | | | |
| <1000 | 29 | 2048 | 0.187 (−0.013 to 0.386), 0.067 | 78.79 | <0.001 | 0.938 |
| ≥1000 | 7 | 768 | 0.097 (−0.051 to 0.245), 0.198 | 0.00 | 0.992 | |
| Types of calcium supplement | | | | | | |

*Table 3 continued on next page*

*Table 3 continued*

| Variable | No. of datasets | No. of participants | BMD difference (95% CI), p value | Heterogeneity between studies | | p value* |
|---|---|---|---|---|---|---|
| | | | | *I²* (%) | p value | |
| Dietary calcium | 17 | 1267 | 0.198 (−0.119 to 0.516), 0.221 | 86.46 | <0.001 | 0.447 |
| Calcium supplementation | 19 | 1519 | 0.129 (0.024 to 0.234), 0.016 | 0.00 | 0.664 | |
| Supplementation with or without vitamin D | | | | | | |
| Without vitamin D | 26 | 2095 | 0.256 (0.056 to 0.456), 0.012 | 78.77 | <0.001 | 0.057 |
| With vitamin D | 10 | 700 | −0.059 (−0.214 to 0.096), 0.456 | 0.00 | 0.608 | |
| Femoral neck | | | | | | |
| Age | | | | | | |
| Prepeak | 13 | 1018 | 0.249 (0.043 to 0.454), 0.018 | 58.27 | 0.004 | <0.001 |
| Peripeak | 2 | 200 | 1.045 (0.701 to 1.390), <0.001 | 0.00 | 0.348 | |
| Duration | | | | | | |
| <18 months | 9 | 648 | 0.569 (0.223 to 0.914), 0.001 | 75.38 | <0.001 | 0.194 |
| ≥18 months | 6 | 570 | 0.107 (−0.062 to 0.276), 0.213 | 0.00 | 0.467 | |
| Sex | | | | | | |
| Women-only trials | 5 | 397 | 0.742 (0.267 to 1.217), 0.002 | 74.47 | 0.004 | 0.129 |
| Trials with men and women | 10 | 793 | 0.195 (−0.027 to 0.418), 0.086 | 57.60 | 0.012 | |
| Regions | | | | | | |
| Asian | 10 | 793 | 0.195 (−0.027 to 0.418), 0.086 | 57.60 | 0.012 | 0.129 |
| Western | 5 | 397 | 0.742 (0.267 to 1.217), 0.002 | 74.47 | 0.004 | |
| Types of calcium supplement | | | | | | |
| Dietary calcium | 9 | 684 | 0.218 (−0.029 to 0.464), 0.083 | 60.89 | 0.009 | 0.367 |
| Calcium supplementation | 6 | 506 | 0.609 (0.162 to 1.056), 0.008 | 78.02 | 0.000 | |

*Table 3 continued on next page*

*Table 3 continued*

| Variable | No. of datasets | No. of participants | BMD difference (95% CI), p value | Heterogeneity between studies | | p value* |
|---|---|---|---|---|---|---|
| | | | | $I^2$ (%) | p value | |
| Supplementation with or without vitamin D | | | | | | |
| Without vitamin D | 5 | 518 | 0.269 (−0.025 to 0.563), 0.073 | 52.38 | 0.078 | 0.865 |
| With vitamin D | 10 | 700 | 0.393 (0.067 to 0.719), 0.018 | 76.45 | <0.001 | |
| Total hip | | | | | | |
| Age | | | | | | |
| Prepeak | 13 | 1194 | 0.273 (−0.150 to 0.696), 0.206 | 91.78 | <0.001 | <0.001 |
| Peripeak | 1 | 121 | −1.936 (−2.346 to −1.525), <0.001 | 0.00 | 1.000 | |
| Duration | | | | | | |
| <18 months | 6 | 542 | −0.226 (−0.514 to 0.061), 0.123 | 61.79 | 0.023 | 0.083 |
| ≥18 months | 8 | 773 | 0.385 (−0.495 to 1.264), 0.392 | 96.76 | <0.001 | |
| Sex | | | | | | |
| Women-only trials | 3 | 420 | −0.202 (−1.851 to 1.448), 0.81 | 98.13 | <0.001 | 0.499 |
| Trials with men and women | 11 | 866 | 0.205 (−0.276 to 0.685), 0.404 | 91.70 | <0.001 | |
| Regions | | | | | | |
| Asian | 10 | 894 | 0.043 (−0.087 to 0.172), 0.516 | 0.00 | 0.691 | 0.914 |
| Western | 4 | 392 | 0.325 (−1.788 to 2.438), 0.763 | 98.71 | <0.001 | |
| Supplementation with or without vitamin D | | | | | | |
| Without vitamin D | 6 | 815 | 0.226 (−0.837 to 1.289), 0.677 | 97.87 | <0.001 | 0.981 |
| With vitamin D | 8 | 500 | 0.032 (−0.144 to 0.208), 0.721 | 0.00 | 0.663 | |
| Total body | | | | | | |
| Age | | | | | | |

*Table 3 continued*

| Variable | No. of datasets | No. of participants | BMD difference (95% CI), p value | Heterogeneity between studies | | p value* |
|---|---|---|---|---|---|---|
| | | | | $I^2$ (%) | p value | |
| Prepeak | 50 | 3762 | 0.168 (0.029 to 0.308), 0.018 | 79.47 | <0.001 | <0.001 |
| Peripeak | 1 | 121 | −0.716 (−1.086 to −0.347), <0.001 | 0.00 | 1.000 | |
| Duration | | | | | | |
| <18 months | 26 | 1760 | 0.146 (−0.095 to 0.387), 0.235 | 83.36 | <0.001 | 0.902 |
| ≥18 months | 25 | 2634 | 0.143 (−0.027 to 0.313), 0.100 | 77.82 | <0.001 | |
| Sex | | | | | | |
| Women-only trials | 23 | 2139 | 0.227 (−0.021 to 0.476), 0.073 | 86.47 | <0.001 | 0.593 |
| Trials with men and women | 28 | 2089 | 0.082 (−0.076 to 0.240), 0.310 | 70.54 | <0.001 | |
| Regions | | | | | | |
| Asian | 22 | 2142 | 0.186 (−0.004 to 0.375), 0.055 | 79.98 | <0.001 | 0.569 |
| Western | 29 | 2086 | 0.120 (−0.094 to 0.334), 0.273 | 81.74 | <0.001 | |
| Baseline calcium intake, mg/day | | | | | | |
| <714 | 30 | 2765 | 0.123 (−0.082 to 0.327), 0.239 | 86.14 | <0.001 | 0.307 |
| ≥714 | 21 | 1463 | 0.186 (0.014 to 0.358), 0.034 | 59.78 | <0.001 | |
| Calcium dose, mg/day | | | | | | |
| <1000 | 37 | 2779 | 0.172 (−0.017 to 0.361), 0.074 | 84.50 | <0.001 | 0.895 |
| ≥1000 | 14 | 1314 | 0.090 (−0.075 to 0.255), 0.283 | 51.43 | 0.013 | |
| Types of calcium supplement | | | | | | |
| Dietary calcium | 26 | 2087 | 0.084 (−0.109 to 0.277), 0.392 | 80.09 | <0.001 | 0.429 |
| Calcium supplementation | 25 | 2141 | 0.215 (0.004 to 0.427), 0.046 | 81.58 | <0.001 | |
| Supplementation with or without vitamin D | | | | | | |

*Table 3 continued on next page*

*Table 3 continued*

| Variable | No. of datasets | No. of participants | BMD difference (95% CI), p value | Heterogeneity between studies | | |
|---|---|---|---|---|---|---|
| | | | | $I^2$ (%) | p value | p value* |
| Without vitamin D | 35 | 2910 | 0.205 (0.017 to 0.393), 0.033 | 83.03 | <0.001 | 0.320 |
| With vitamin D | 15 | 1388 | 0.030 (−0.188 to 0.249), 0.786 | 75.35 | <0.001 | |

*p value for heterogeneity between subgroups.

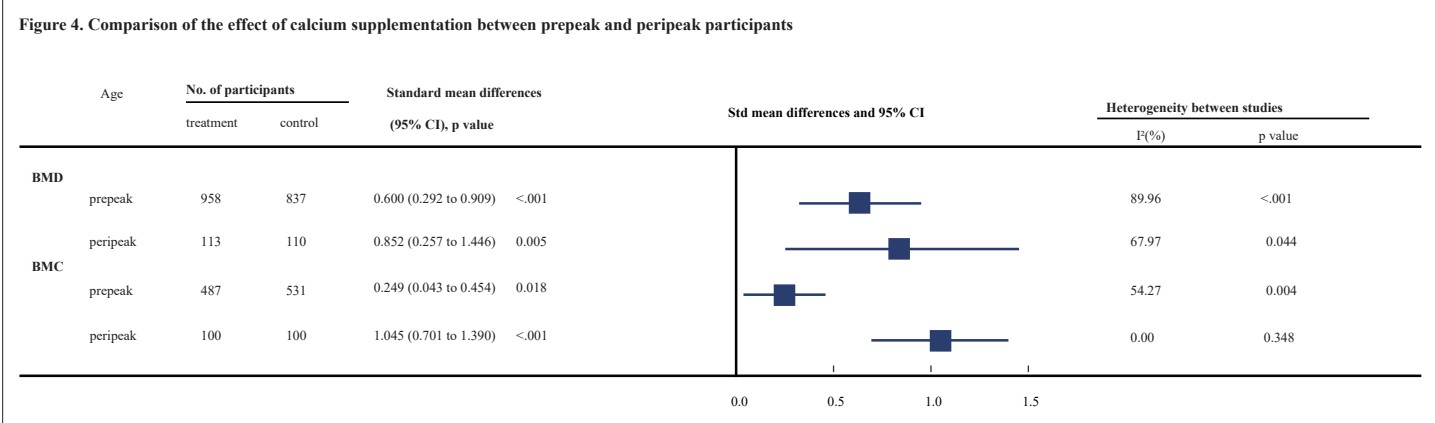

**Figure 4.** Comparison of the effect of calcium supplementation between prepeak and peripeak participants.

0.002, respectively) than in trials including men and women (0.556, 95% CI: 0.233–0.879, p = 0.001; 0.195, 95% CI: −0.027 to 0.418, p = 0.086).

When considering the sources of participants, the improvement effects on femoral neck and total body BMD or on femoral neck and lumbar spine BMC were obviously stronger in Western countries than in Asian countries.

Subgroup analyses by the level of dietary calcium intake at baseline showed that, for femoral neck BMD, the beneficial effect was significant only in the lower subgroup receiving <714 mg/day (0.581, 95% CI: 0.266–0.896; p < 0.001); for total body BMD, the beneficial effect was slightly greater in the lower subgroup receiving <714 mg/day (0.363, 95% CI: 0.127–0.599; p = 0.003); for total hip BMD and lumbar spine BMC, however, the beneficial effects were statistically significant in the higher subgroup receiving ≥714 mg/day (0.723, 95% CI: 0.245–1.201; p = 0.003 and 0.2, 95% CI: 0.052–0.348; p = 0.008, respectively).

Subgroup analyses based on calcium supplement dosages demonstrated a statistically significant effect on femoral neck and total body BMD in the lower dose subgroup receiving <1000 mg/day (0.717, 95% CI: 0.349–1.085; p < 0.001 and 0.392, 95% CI: 0.161–0.624; p = 0.001, respectively) but not in the higher dose subgroup receiving ≥1000 mg/day.

When considering the different sources of calcium, both calcium sources from dietary intake and additional calcium supplements exerted significantly positive effects on femoral neck BMD (0.728, 95% CI: 0.311–1.144, p < 0.001; 0.510, 95% CI: 0.101–0.919, p = 0.014) and total body BMD (0.290, 95% CI: 0.054–0.526, p = 0.016; 0.405, 95% CI: 0.195–0.615, p < 0.001). For BMCs of the lumbar spine and femoral neck, only calcium supplements other than dietary intake had a significant improvement effect.

To explore the longevity of the beneficial effect, we performed subgroup analyses and found that calcium supplementation improved the BMD levels during the follow-up periods after the end of intervention, and the beneficial effect was maintained for at least 1 year after the intervention (0.933, 95% CI: 0.323–1.664, p = 0.004). However, this beneficial effect seemed to disappear when the follow-up period exceeded 2 years.

In order to compare the effect of the presence or absence of vitamin D on the effect of calcium supplementation, we divided all the data into two groups and ran the calculations separately. Calcium supplementation with vitamin D showed greater beneficial effects on femoral neck BMD and BMC (0.758, 95% CI: 0.350–1.166, p < 0.001; 0.393, 95% CI: 0.067–0.719, p = 0.018). However, for BMCs of lumbar spine and total body, as well as total body BMD, only calcium supplementation without vitamin D had a significant improvement effect.

## Sensitivity analysis

Sensitivity analyses including only trials with a low risk of bias (high quality, see *Supplementary file 4*) showed that the improvement effects on femoral neck BMD and BMC remained statistically significant and stable (0.356, 95% CI: 0.064–0.648, p = 0.017; 0.249, 95% CI: 0.043–0.454, p = 0.018). The result for total body BMD was also stable (0.343, 95% CI: 0.098–0.588, p = 0.006). However,

for lumbar spine and total body BMCs, the positive effect was not statistically significant. For other sites, the results were generally consistent with those of the primary analyses. Additional sensitivity analyses using fixed-effect models (see *Supplementary file 5*), performing cumulative meta-analysis (see *Supplementary file 6*), and excluding studies had been included in previous meta-analysis (see *Supplementary file 7*) showed generally consistent results with the primary analyses.

### GRADE scoring

*Supplementary file 8* shows a summary of the GRADE assessments of the overall certainty of the evidence for the effect of calcium supplementation on bone measurements. The evidence was graded as moderate for all sites. All of these outcomes were downgraded for inconsistency. For femoral neck BMD, it was downgraded because of strongly suspected publication bias, however, it was upgraded due to the effect size was over 0.5. In summary, the outcome of femoral neck BMD was graded as moderate.

### Heterogeneity analysis

In general, the heterogeneity between trials was obvious in the analysis for BMD ($p < 0.001$, $I^2 = 86.28\%$) and slightly smaller for BMC ($p < 0.001$, $I^2 = 79.28\%$). The intertrial heterogeneity was significantly distinct across the sites measured. Subgroup analyses and meta-regression analyses suggested that this heterogeneity could be explained partially by differences in age, duration, calcium dosages, types of calcium supplement, supplementation with or without vitamin D, baseline calcium intake levels, sex, and region of participants (*Tables 2 and 3* and *Supplementary file 9*).

### Publication bias

Funnel plots, Begg's rank correlation, and Egger's regression test for each outcome bias are presented in *Supplementary file 10*. Publication bias was obvious in the femoral neck BMD. The adjusted effect size analyzed using the trim and fill method also showed a difference from the unadjusted value. Except for the outcome above, no evidence for publication bias was found. The adjusted summary effect size analyzed using the trim and fill method did not show substantial changes as well, which also implies no evidence of publication bias.

## Discussion

This meta-analysis comprehensively summarized the evidence for the efficiency of calcium supplementation in young people before the peak of bone mass and at the plateau period. The findings indicated significant improvement effects of calcium supplements on both BMD and BMC, especially on the femoral neck.

Numerous recent systematic reviews have concluded that there is no evidence for associations between calcium supplements and reduced risk of fracture or improvement of bone density in people aged over 50 years (*Tai et al., 2015*; *Zhao et al., 2017*; *Bolland et al., 2015*; *Bristow et al., 2022*). Since calcium supplements are unlikely to translate into clinically meaningful reductions in fractures or improvement of bone mass in aged people, we wondered if it is possible to increase bone mass at the peak by administering calcium supplements before the age of reaching the PBM or at the plateau of this peak to prevent osteoporosis and reduce the risk of fractures in later life. To the best of our knowledge, this is the first meta-analysis to focus on age before achieving PBM or age at the plateau of PBM, at which the risk of fracture is extremely low. Why did we do such a meta-analysis? Instead of traditionally solving problems when they occurred, that is, treating osteoporosis after a patient has developed osteoporosis, our research attempted to explore the effects of preventive intervention before reaching the plateau and before osteoporosis development. Our study suggests that calcium supplementation can significantly boost peak bone content, which can improve bone mass. Since calcium supplementation in elderly individuals occurs late and has no influence, our findings have critical implications for the early prevention of fractures in the elderly population and provide better insights for the current situation of calcium supplementation. Preventive calcium supplementation in young populations is a shift in the window of intervention for osteoporosis, not limited to a certain age group but involving the whole life cycle of bone health.

Is there any difference in supplementation of calcium before or after the achievement of the PBM? We found that calcium supplementation improved the bone mass at the femoral neck in both the prepeak and peripeak subjects; furthermore, it is worth noting that the improvement effect was obviously stronger in the peripeak population (≥20–35 years) than in the prepeak population (<20 years). Based on our findings and the negative associations of calcium supplements with bone outcomes in aged people from previous studies, one can conclude that young adulthood may be the best intervention window to optimize bone mass, especially the PBM; moreover, our study indicates the importance of calcium supplementation at this age instead of the often-mentioned age groups of children or elderly individuals. The findings of our study provide completely new insight into a novel intervention window in young adulthood to improve bone mass and further prevent osteoporosis and fractures in their late lifespan. To synthesize previously published studies in children, we found a meta-analysis conducted by *Winzenberg et al., 2006* that included 19 studies involving 2859 children and found a small effect on total body BMC and no effect on lumbar spine BMD in children, which was in line with our finding. However, they found no effect on BMD at the femoral neck, which was inconsistent with our result. We therefore performed a sensitivity analysis, excluding all the literature they included, and found that the results of our newly included studies, 28 in total, were generally consistent with the primary results. We also performed a sensitivity analysis incorporating only the studies they pooled and found a statistically significant effect for BMD in the femoral neck and total body, while the results for total body BMC were nonsignificant (see e *Supplementary file 7*). These slightly different findings can be interpreted as follows: first, we included more and updated literature; second, they used only endpoint data directly, whereas we used change data, taking into account the difference in baseline conditions; third, we used change data to represent the change before and after calcium supplementation more directly. Another meta-analysis conducted by *Huncharek et al., 2008* included 21 studies involving 3821 subjects and pooled three reports involving subjects with low baseline calcium intake and reported a statistically significant summary of the mean BMC in children. Combining the above published literature with our conclusions, it can be concluded that calcium supplementation is more effective in young adults aged 20–35 years than in children. Although this issue needs to be confirmed in the future, our findings highlight the importance of this intervention window of approximately 10–15 years at the peri-PBM period, which is better than the pre-PBM period.

To explore whether there is a difference between dietary calcium intake and calcium supplements, our subgroup analyses suggested that one can obtain this beneficial effect from both calcium sources, including dietary intake and calcium supplements. For BMD at the femoral neck, dietary calcium seemed to exert a better effect than calcium supplements. Similarly, we also found that the improvement effect was statistically significant only in subjects supplied with calcium dosages lower than 1000 mg/day. These findings support the hypothesis that there may be a threshold dose of calcium supplementation; when exceeded, the effect does not increase. Our findings are consistent with the previous research by *Prentice, 2002*, which is that no additional benefit is associated with an intake above the currently recommended dose at the population level. The underlying mechanisms are unclear and need to be elucidated in future studies.

To explore whether the effect of improving BMD or BMC is due to calcium alone or calcium plus vitamin D, our subgroup analyses found that calcium supplementation with vitamin D had greater beneficial effects on both the femoral neck BMD and BMC than calcium supplementation without vitamin D. However, for both BMD and BMC at the other sites (including lumbar spine, total hip, and total body), the observed effects in the subgroup without vitamin D supplementation appeared to be slightly better than in the subgroup with vitamin D supplementation. Therefore, these results suggested that calcium supplementation alone could improve BMD or BMC, although additional vitamin D supplementation may be beneficial in improving BMD or BMC at the femoral neck.

To determine the differences between high dietary intake and low dietary intake of calcium at baseline, our subgroup analyses showed that the improvement effect seemed to be stronger in subjects with high intake at baseline than in those in the lower subgroup. Interestingly, these results were in accordance with the findings of subgroup analyses by population area, which suggested that calcium supplementation was more effective in Western populations, whose level of baseline calcium intake is normally higher than that in Asian countries. However, these findings are likely to be contrary to our common sense, which is, that under normal circumstances, the effects of calcium supplementation

should be more obvious in people with lower calcium intake than in those with higher calcium intake. Therefore, this issue needs to be tested and confirmed in future trials.

To investigate changes in the effect of calcium supplementation after cessation, our subgroup analysis showed that the effect remained significant 1 year after cessation, particularly at various sites of BMD. For studies with a follow-up period longer than 1 year, we included only two articles: one study *Lambert et al., 2008* with 2 years of follow-up after calcium supplementation was stopped and another study *Chevalley et al., 2005b* with 7 years of follow-up. Their results were pooled and showed that the effects of calcium supplementation no longer persisted. The number of studies is too small for us to explore how long the effects of calcium supplementation will last, and well-designed cohort studies are needed in the future. In the meantime, we have found a point to ponder about whether gains can be made when calcium supplementation is restarted after a period of withdrawal and what other changes in the organism remain to be discovered.

Several limitations need to be considered. First, there was substantial intertrial heterogeneity in the present analysis, which might be attributed to the differences in baseline calcium intake levels, regions, age, duration, calcium dosages, types of calcium supplement, supplementation with or without vitamin D and sexes according to subgroup and meta-regression analyses. To take heterogeneity into account, we used random-effect models to summarize the effect estimates, which could reduce the impact of heterogeneity on the results to some extent. Second, our research failed to clearly compare the difference between males and females due to the limitation of existing data – some studies provided merged data of males and females without males alone. Based on the existing data, the beneficial effect was more obvious when subjects were limited to women only, which needs to be validated in future trials. Third, we found that few of the existing studies focused on the 20- to 35-year age group, which was why there were only three studies of this age group that met our inclusion criteria; although the number was small, our evidence was of high quality, and the results were stable, especially in the femoral neck. We also tried to find mechanisms related to bone metabolism in the age group of 20–35 years, but few studies have focused on this age group; most studies have focused only on mechanisms related to older people or children. Therefore, more high-quality RCTs and studies on the exploration of mechanisms focusing on the 20- to 35-year age group are needed in the future. Finally, as some of the studies did not provide the physical activity levels of the participants, we failed to exclude the effect of physical activity on the results.

This study has several strengths. In this first systematic review by meta-analysis to focus on people at the age before achieving PBM and at the age around the peak of bone mass, we comprehensively searched for all of the currently eligible trials and included a total of 7382 participants (including 3283 calcium supplement users and 4099 controls), which added reliability to our findings. Another strength is the high consistency of the results across predesigned subgroup analyses and sensitivity analyses. Additionally, we analyzed both BMD and BMC separately for the different measurement sites rather than using the mean of all combined values to draw conclusions, which has the advantage of obtaining changes in bone indexes at different sites and drawing more accurate conclusions.

In conclusion, calcium supplementation can significantly improve BMD and BMC, especially at the femoral neck. Moreover, supplementation in people who are at the plateau of their PBM has a better effect. Although further well-designed RCTs with larger sample sizes are required to verify our findings, we provide a new train of thought regarding calcium supplementation and the evaluation of its effects. In terms of bone health and the full life cycle of a person, the intervention window of calcium supplementation should be advanced to the age around the plateau of PBM, namely, at 20–35 years of age.

## Acknowledgements

We thank all authors and participants of the original studies. We also thank American Journal Experts for English language polishing of the manuscript.

# Additional information

### Funding

| Funder | Grant reference number | Author |
|---|---|---|
| Wenzhou Medical University | 89219029 | Shuran Wang |

The funders had no role in study design, data collection, and interpretation, or the decision to submit the work for publication.

### Author contributions

Yupeng Liu, Conceptualization, Formal analysis, Supervision, Writing - review and editing; Siyu Le, Software, Formal analysis, Methodology, Writing - original draft; Yi Liu, Supervision, Validation, Writing - original draft; Huinan Jiang, Binye Ruan, Yufeng Huang, Xuemei Ao, Xudong Shi, Writing - original draft; Xiaoyi Fu, Shuran Wang, Conceptualization, Supervision, Writing - review and editing

### Author ORCIDs

Yupeng Liu ⓘ http://orcid.org/0000-0002-8652-1987
Yi Liu ⓘ http://orcid.org/0000-0003-2135-6594
Shuran Wang ⓘ http://orcid.org/0000-0002-6480-1195

### Decision letter and Author response

Decision letter https://doi.org/10.7554/eLife.79002.sa1
Author response https://doi.org/10.7554/eLife.79002.sa2

# Additional files

### Supplementary files

- Supplementary file 1. Search strategies.

- Supplementary file 2. Excluded trials and reasons for exclusion.

- Supplementary file 3. Risk-of-bias assessment for eligible trials.

- Supplementary file 4. Sensitivity analyses excluding studies of low or medium quality. (A) Sensitivity analyses excluding studies of low or medium quality in bone mineral density (BMD). (B) Sensitivity analyses excluding studies of low or medium quality in bone mineral content (BMC).

- Supplementary file 5. Sensitivity analysis by comparisons of fixed and random-effect models. (A) Sensitivity analysis by comparisons of fixed and random-effect models for bone mineral density (BMD). (B) Sensitivity analysis by comparisons of fixed and random-effect models for bone mineral content (BMC).

- Supplementary file 6. Cumulative meta-analysis according to sample size. (A) Cumulative meta-analysis according to sample size in lumbar spine bone mineral density (LSBMD). (B) Cumulative meta-analysis according to sample size in femoral neck bone mineral density (FNBMD). (C) Cumulative meta-analysis according to sample size in total hip bone mineral density (THBMD). (D) Cumulative meta-analysis according to sample size in total body bone mineral density (TBBMD). (E) Cumulative meta-analysis according to sample size in lumbar spine bone mineral content (LSBMC). (F) Cumulative meta-analysis according to sample size in femoral neck bone mineral content (FNBMC). (G) Cumulative meta-analysis according to sample size in total hip bone mineral content (THBMC). (H) Cumulative meta-analysis according to sample size in total body bone mineral content (TBBMC).

- Supplementary file 7. Sensitivity analyses by comparisons of the pooled results of the trials included in previous study and trials newly added in our current study. (A) Sensitivity analyses by comparisons of the pooled results of the trials included in previous study and trials newly added in our current study of bone mineral density (BMD). (B) Sensitivity analyses by comparisons of the pooled results of the trials included in previous study and trials newly added in our current study of bone mineral content (BMC).

- Supplementary file 8. GRADE assessment.

- Supplementary file 9. Meta-regression for age, region, Ca dosage, baseline intake and sample size

on bone mineral density (BMD) and bone mineral content (BMC).
- Supplementary file 10. Publication bias.
- MDAR checklist
- Reporting standard 1. PRISMA checklist.

### Data availability

All data in this analysis are based on published studies. Source Data files have been provided for Figures 2 and 3. Figure 2–Source Data 1–4 and Figure 3–Source Data 1–4 contain the numerical data used to generate the figures. Supplementary data files contain all raw tabulated data are provided in appendix.

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
