## [Editor Report]

This manuscript is of interest to researchers and practitioners who are researching or treating osteoporosis. The effect of calcium supplementation on bone mineral density improvement, which was not shown in the elderly or children was shown in subjects younger than 35 years of age near PBM. It provides an important conclusion that calcium supplementation should be seriously considered at that age to prevent osteoporosis.

---

## [Decision Letter]

**Decision letter after peer review:**

Thank you for submitting your article "The effect of calcium supplementation in people under 35 years old: A systematic review and meta-analysis of randomized controlled trials" for consideration by *eLife*. Your article has been reviewed by 3 peer reviewers, one of whom is a member of our Board of Reviewing Editors, and the evaluation has been overseen by Mone Zaidi as the Senior Editor. The reviewers have opted to remain anonymous.

Essential revisions:

(1) Please describe the analysis software and its version used in the Methods.

(2) Please provide information regarding vitamin D supplementation and perform a sub-analysis to see if there are any differences with or without vitamin D supplementation in the results.

(3) Please correct the first paragraph of the discussion. Comments on femoral neck anatomy, fracture, and mortality in the elderly do not seem to fit the point of this manuscript.

(4) I just suggest making a meta-PCA, as demonstrated before in this paper: https://www.nature.com/articles/ncomms13357.

*Reviewer #1 (Recommendations for the authors):*

The outcomes and clinical lessons are very clear and the methodology to draw conclusions is appropriate. The strengths and the limitations of this systematic review are clearly outlined.

The biggest limitation of this study is that the heterogeneity is large and the random effect model is used appropriately.

It would be better to describe the analysis software used in the Methods.

*Reviewer #2 (Recommendations for the authors):*

I recommend that the authors provide information regarding vitamin D supplementation and perform a sub-analysis to see if there are any differences with or without vitamin D supplementation in the results.

Please correct the first paragraph of the discussion. Comments on femoral neck anatomy, fracture, and mortality in the elderly do not seem to fit the point of this manuscript.

*Reviewer #3 (Recommendations for the authors):*

In order to elevate these results, I just suggest making a meta-PCA, as demonstrated before in this paper: https://www.nature.com/articles/ncomms13357

Just consider if it's not time-consuming.

---

## [Author Response]

Essential revisions:(1) Please describe the analysis software and its version used in the Methods.

Thank you for your careful review. According to your suggestions, we have described the information in details in the methods section, as follows:

Data extraction and integration were done on Microsoft Office Excel (version 2011). Meta-analysis, subgroup analysis and sensitivity analysis were all performed by Comprehensive Meta Analysis (version 3.3.070, Biostat, Englewood, NJ). (See Lines 144-147 on Page 7 in the Main Text)

2) Please provide information regarding vitamin D supplementation and perform a sub-analysis to see if there are any differences with or without vitamin D supplementation in the results.

Thank you very much for your comments and suggestions. According to your suggestions, we have added the corresponding analyses regarding calcium supplementation with or without vitamin D supplementation. Among the included RCTs, 32 trials used calcium-only supplementation (without vitamin D supplementation) and 11 trials used calcium plus vitamin D supplementation. The detailed information are provided in the Author response table 1 and Author response table 2. We have added subgroup analyses by vitamin D supplementation as you suggested, and the corresponding results are provided in Author response table 3 and Author response table 4.

**Author response table 1. sa2table1:** Information regarding vitamin D supplementation in the included trials.

Subgroups	No. of trials	Trials
Without vitamin D supplementation	32	Johnston 1992; Lee 1994; Lloyd 1994; Lee 1995; Lloyd 1996; Bonjour 1997; Cadogan 1997; Nowson 1997; Sandra 2003; Rozen 2003; Volek 2003; Specker 2003; Stear 2003; Molgaard 2004; Cameron 2004; Winters-Stone 2004; Lau 2004; Gibbons 2004; Chevalley 2005; Matkovic 2005; Barger-Lux 2005; Prentice 2005; Courteix 2005; Ho 2005; Chevalley 2005; Ward 2007; Bass 2007; Lambert 2008; Yin 2010; Ekbote 2011; Ward 2014; Vogel 2017;
With vitamin D supplementation	11	Moyer-Mileur 2003; Du 2004; Cheng 2005; Zhu 2008; Hemayattalab 2010; Islam 2010; Khadilkar 2012; Arab 2012; Ma 2014; Zhang 2014; Lu 2019;

**Author response table 2. sa2table2:** Detailed information regarding vitamin D supplementation in the trials with vitamin D supplementation.

Trials	Vitamin D Supplementation	
		
Moyer-Mileur 2003	Arm 1: 800 mg calcium carbonate and vitamin D (400 IU);Arm 2: placebo	
Du 2004	Arm 1: milk;Arm 2: milK^+^ vitamin D (5 or 8mg cholecalciferol);Arm 3: control	
Cheng 2005	Arm 1: calcium (1000 mg) vitamin D3 (200 IU)Arm 2: calcium (1000 mg),Arm 3: cheese (1000 mg calcium)Arm 4: placebo	
Zhu 2008	Arm 1: calcium fortified milk (Ca milk)Arm 2: calcium and vitamin D fortified milk (CaD milk)Arm 3: control	
Hemayattalab 2010	Arm 1: calcium plus vitamin DArm 2: control	
Islam 2010	Arm 1 – Arm 3: 600 mg of calcium with 10mg of VD; 600 mg of calcium plus multiple micronutrients with 10mg of VDArm 4: control	
Khadilkar 2012	Arm 1: calcium with 30,000 IU vitamin D3 (cholecalciferol);Arm 2: calcium and multivitamin plus zinc with 30,000 IU vitamin D3Arm 3: control	
Arab 2012	Arm 1: calcium plus vitamin DArm 2: control	
Ma 2014	Arm 1 – Arm 3: low, medium and high doses of calcium plus 200 IU vitamin DArm 4: control	
Zhang 2014	Arm 1 – Arm 3: low, medium and high doses of calcium plus 200 IU vitamin DArm 4: control	
Lu 2019	Arm 1 – Arm 3: milk powder fortified with vitamin D 400 IU plus calcium 300, or 600, or 900 mgArm 4: control	

**Author response table 3. sa2table3:** Subgroup analyses by calcium supplementation with or without vitamin D on bone mineral density at lumbar spine, femoral neck, total hip and total body.

Subgroups	No. of datasets	No. of participants	Mean difference(95% CI),P-value	Heterogeneity between studies	P-value for difference between subgroups	
				I² (%)	P-value	
Lumbar Spine						
without vitamin D	22	2520	0.140 (-0.047 to 0.327), 0.143	78.59	<0.001	0.468
with vitamin D	13	1078	0.008 (-0.160 to 0.176), 0.926	44.69	0.041	
Femoral Neck						
without vitamin D	10	1331	0.477 (0.045 to 0.910), 0.031	91.44	<0.001	0.119
with vitamin D	14	794	0.758 (0.350 to 1.166), <0.001	85.38	<0.001	
Total Hip						
without vitamin D	7	894	0.506 (-0.138 to 1.149), 0.123	94.78	<0.001	0.546
with vitamin D	11	878	0.101 (-0.191 to 0.393), 0.498	78.22	<0.001	
Total Body						
without vitamin D	22	2657	0.701 (0.327 to 1.076), <0.001	94.83	<0.001	0.137
with vitamin D	15	1625	0.156 (-0.156 to 0.468), 0.327	88.94	<0.001	

**Author response table 4. sa2table4:** Subgroup analyses by calcium supplementation with or without vitamin D on bone mineral content at lumbar spine, femoral neck, total hip and total body.

Subgroups	No. of datasets	No. of participants	Mean difference (95% CI),P-value	Heterogeneity between studies	P-value for difference between subgroups	
				I² (%)	P-value	
Lumbar Spine						
without vitamin D	26	2095	0.256 (0.056 to 0.456), 0.012	78.77	<0.001	0.057
with vitamin D	10	700	-0.059 (-0.214 to 0.096), 0.456	0.00	0.608	
Femoral Neck						
without vitamin D	5	518	0.269 (-0.025 to 0.563), 0.073	52.38	0.078	0.865
with vitamin D	10	700	0.393 (0.067 to 0.719), 0.018	76.45	<0.001	
Total Hip						
without vitamin D	6	815	0.226 (-0.837 to 1.289), 0.677	97.87	<0.001	0.981
with vitamin D	8	500	0.032 (-0.144 to 0.208), 0.721	0.00	0.663	
Total Body						
without vitamin D	35	2910	0.205 (0.017 to 0.393), 0.033	83.03	<0.001	0.320
with vitamin D	15	1388	0.030 (-0.188 to 0.249), 0.786	75.35	<0.001	

When we pooled the data from the two subgroups separately, we found that calcium supplementation with vitamin D had greater beneficial effects on both the femoral neck BMD (MD: 0.758, 95% CI: 0.350 to 1.166, P < 0.001 VS. MD: 0.477, 95% CI: 0.045 to 0.910, P = 0.031) and the femoral neck BMC (MD: 0.393, 95% CI: 0.067 to 0.719, P = 0.018 VS. MD: 0.269, 95% CI: -0.025 to 0.563, P = 0.073) than calcium supplementation without vitamin D. However, for both BMD and BMC at the other sites (including lumbar spine, total hip, and total body), the observed effects in the subgroup without vitamin D supplementation appeared to be slightly better than in the subgroup with vitamin D supplementation.

We have added the relevant parts in the sections of Method, Results and Discussion in the main text of the revised manuscript, as follows:

Method: We further conducted some post hoc subgroup analyses according to the level of calcium intake at baseline (< 714 vs. ≥ 714 mg/day, based on the median value), the calcium supplementation dose (<1000 versus ≥1000 mg/day, based on the median value) and vitamin D supplementation (with or without vitamin D). (See Lines 128-131 on Page 6-7)

Results: In order to compare the effect of the presence or absence of vitamin D on the effect of calcium supplementation, we divided all the data into two groups and ran the calculations separately. Calcium supplementation with vitamin D showed greater beneficial effects on femoral neck BMD and BMC (0.758, 95% CI: 0.350 to 1.166, P<.001; 0.393, 95% CI: 0.067 to 0.719, P=0.018). However, for BMCs of lumbar spine and total body, as well as total body BMD, only calcium supplementation without vitamin D had a significant improvement effect. (see Lines 260-265 on Page 13)

Discussion: To explore whether the effect of improving BMD or BMC is due to calcium alone or calcium plus vitamin D, our subgroup analyses found that calcium supplementation with vitamin D had greater beneficial effects on both the femoral neck BMD and BMC than calcium supplementation without vitamin D. However, for both BMD and BMC at the other sites (including lumbar spine, total hip, and total body), the observed effects in the subgroup without vitamin D supplementation appeared to be slightly better than in the subgroup with vitamin D supplementation. Therefore, these results suggested that calcium supplementation alone could improve BMD or BMC, although additional vitamin D supplementation may be beneficial in improving BMD or BMC at the femoral neck. (see Lines 369-376 on Page 23)

(3) Please correct the first paragraph of the discussion. Comments on femoral neck anatomy, fracture, and mortality in the elderly do not seem to fit the point of this manuscript.

Thank you for your kind comments. According to your suggestions, we have removed these sentences about femoral neck anatomy, fracture and mortality in the elderly, which are not applicable to be mentioned in the first paragraph of the discussion. The first paragraph of the revised discussion is showed as follows:

This meta-analysis comprehensively summarized the evidence for the efficiency of calcium supplementation in young people before the peak of bone mass and at the plateau period. The findings indicated significant improvement effects of calcium supplements on both BMD and BMC, especially on the femoral neck. (See Lines 309-312 on Page 21 in the Main Text)

Please tell us directly if it needs further corrections, we will be very grateful and truly appreciate it, and try our best to revise it.

(4) I just suggest making a meta-PCA, as demonstrated before in this paper: https://www.nature.com/articles/ncomms13357.

Thank you very much for your excellent and thoughtful suggestions. Following your suggestion, we have thoroughly and carefully read your recommended paper and some related papers using meta-PCA (such as doi: 10.1002/14651858.CD013229.pub2; doi: 10.1016/j.numecd.2022.03.010; doi: 10.1371/journal.pcbi.1005890; doi: 10.1093/bioinformatics/btx765. etc.). Janina S Ried and Janina M Jeff et al. have developed this PCA-based meta-analysis approach to capture variations in multiple traits simultaneously across multiple studies in a uniform way. This approach can be applied to discover what single-trait analysis cannot capture.

We must admit that this is the first time we have heard and learned about this method, which will be of great help to us in our future research. However, after careful discussions among all members of our research team, we were unable to make good use of the meta-PCA method in our current study to analyse the available data in a short period of time. We apologise very sincerely for this and hope that you would understand our plight.

Thanks again for your insightful and thoughtful comments. We have made our best efforts to implement all your suggestions in the revised manuscript. However, there may remain some deficiencies. Please tell us directly if it needs further corrections, we will be very grateful and appreciate it, and try our best to revise this manuscript.

Reviewer #1 (Recommendations for the authors):The outcomes and clinical lessons are very clear and the methodology to draw conclusions is appropriate. The strengths and the limitations of this systematic review are clearly outlined.The biggest limitation of this study is that the heterogeneity is large and the random effect model is used appropriately.

Thank you for your kind comments. We are also aware of the large inter-trial heterogeneity and have attempted to compensate for this. As you mentioned, the biggest limitation of this study was indeed the large intertrial heterogeneity. Because of this, we have compulsorily chosen the random-effects model, which, as you said, would have been better suited to obtaining more conservative results. In addition, we performed meta-regression and meta-subgroup analyses in order to explore possible sources of the heterogeneity and eventually found that the observed heterogeneity may be due to the differences in age, sexes, regions of subjects, doses, intervention duration, and types of calcium supplementation, dietary calcium intake levels at baseline, and with or without vitamin D supplementation.

We have updated the results and discussions about potential sources of heterogeneity in the revised manuscript, as follows:

In general, the heterogeneity between trials was obvious in the analysis for BMD (P < 0.001, I^2^ = 86.28%) and slightly smaller for BMC (P < 0.001, I^2^ = 79.28%). The intertrial heterogeneity was significantly distinct across the sites measured. Subgroup analyses and meta-regression analyses suggested that this heterogeneity could be explained partially by differences in age, duration, calcium doses, types of calcium supplement, with or without vitamin D supplementation, baseline dietary calcium intake levels, sexes and regions of participants. (See Lines 295-300 on Page 20 in the Main Text)

Several limitations need to be considered. First, there was substantial intertrial heterogeneity in the present analysis, which might be attributed to the differences in dietary calcium intake at baseline, population regions, age, intervention duration, calcium doses, types of calcium supplements, with or without vitamin D supplementation and sexes according to subgroup and meta-regression analyses. To take heterogeneity into account, we used random effect models to summarize the effect estimates, which could reduce the impact of heterogeneity on the results to some extent. (See Lines 396-401 on Page 24 in the Main Text)

It would be better to describe the analysis software used in the Methods.

Thank you very much for your careful review. According to your suggestions, we have described the information in detail in the methods section, as follows:

Data extraction and integration were done on Microsoft Office Excel (version 2011). Meta-analysis, subgroup analysis and sensitivity analysis were all performed by Comprehensive Meta Analysis (version 3.3.070, Biostat, Englewood, NJ). (See Lines 144-147 on Page 7 in the Main Text)

Thanks again for your great patience and considerate suggestions.

These insightful and thoughtful comments really did help us to improve the quality and clarity of this manuscript. However, there may remain some deficiencies. Please tell us directly if it needs further corrections, we will be very grateful and appreciate it, and try our best to revise this manuscript.

Reviewer #2 (Recommendations for the authors):I recommend that the authors provide information regarding vitamin D supplementation and perform a sub-analysis to see if there are any differences with or without vitamin D supplementation in the results.

Thanks for your insightful and thoughtful comments which really did help us to improve the quality and clarity of this manuscript. According to your suggestions, we have added the corresponding analyses regarding calcium supplementation with or without vitamin D supplementation. Among the included RCTs, 32 trials used calcium-only supplementation (without vitamin D supplementation) and 11 trials used calcium plus vitamin D supplementation. The detailed informations are provided in the Author response table 1 and Author response table 2. We have added subgroup analyses by vitamin D supplementation as you suggested, and the corresponding results are provided in Author response table 3 and Author response table 4.

When we pooled the data from the two subgroups separately, we found that calcium supplementation with vitamin D had greater beneficial effects on both the femoral neck BMD (MD: 0.758, 95% CI: 0.350 to 1.166, P < 0.001 VS. MD: 0.477, 95% CI: 0.045 to 0.910, P = 0.031) and the femoral neck BMC (MD: 0.393, 95% CI: 0.067 to 0.719, P = 0.018 VS. MD: 0.269, 95% CI: -0.025 to 0.563, P = 0.073) than calcium supplementation without vitamin D. However, for both BMD and BMC at the other sites (including lumbar spine, total hip, and total body), the observed effects in the subgroup without vitamin D supplementation appeared to be slightly better than in the subgroup with vitamin D supplementation.

We have added the relevant parts in the sections of Method, Results and Discussion in the main text of the revised manuscript, as follows:

Method: We further conducted some post hoc subgroup analyses according to the level of calcium intake at baseline (< 714 vs. ≥ 714 mg/day, based on the median value), the calcium supplementation dose (<1000 versus ≥1000 mg/day, based on the median value) and vitamin D supplementation (with or without vitamin D). (See Lines 128-131 on Page 6-7)

Results: In order to compare the effect of the presence or absence of vitamin D on the effect of calcium supplementation, we divided all the data into two groups and ran the calculations separately. Calcium supplementation with vitamin D showed greater beneficial effects on femoral neck BMD and BMC (0.758, 95% CI: 0.350 to 1.166, P<.001; 0.393, 95% CI: 0.067 to 0.719, P=0.018). However, for BMCs of lumbar spine and total body, as well as total body BMD, only calcium supplementation without vitamin D had a significant improvement effect. (see Lines 260-265 on Page 13)

Discussion: To explore whether the effect of improving BMD or BMC is due to calcium alone or calcium plus vitamin D, our subgroup analyses found that calcium supplementation with vitamin D had greater beneficial effects on both the femoral neck BMD and BMC than calcium supplementation without vitamin D. However, for both BMD and BMC at the other sites (including lumbar spine, total hip, and total body), the observed effects in the subgroup without vitamin D supplementation appeared to be slightly better than in the subgroup with vitamin D supplementation. Therefore, these results suggested that calcium supplementation alone could improve BMD or BMC, although additional vitamin D supplementation may be beneficial in improving BMD or BMC at the femoral neck. (see Lines 369-376 on Page 23)

Please correct the first paragraph of the discussion. Comments on femoral neck anatomy, fracture, and mortality in the elderly do not seem to fit the point of this manuscript.

Thank you for your suggestion. As you mentioned, we have corrected the first paragraph of the discussion and deleted relevant parts about femoral neck anatomy, fracture and mortality in the elderly. The first paragraph of the revised discussion is showed as follows:

This meta-analysis comprehensively summarized the evidence for the efficiency of calcium supplementation in young people before the peak of bone mass and at the plateau period. The findings indicated significant improvement effects of calcium supplements on both BMD and BMC, especially on the femoral neck. (See Lines 309-312 on Page 21 in the Main Text)

Again, thanks a lot for your great patience and considerate suggestions. Your insightful and thoughtful comments really did help us to improve the quality and clarity of this manuscript.

Reviewer #3 (Recommendations for the authors):In order to elevate these results, I just suggest making a meta-PCA, as demonstrated before in this paper: https://www.nature.com/articles/ncomms13357Just consider if it's not time-consuming.

Thank you for your suggestion. As you mentioned, we have corrected the first paragraph of the discussion and deleted relevant parts about femoral neck anatomy, fracture and mortality in the elderly. The first paragraph of the revised discussion is showed as follows:

This meta-analysis comprehensively summarized the evidence for the efficiency of calcium supplementation in young people before the peak of bone mass and at the plateau period. The findings indicated significant improvement effects of calcium supplements on both BMD and BMC, especially on the femoral neck. (See Lines 309-312 on Page 21 in the Main Text)

Again, thanks a lot for your great patience and considerate suggestions. Your insightful and thoughtful comments really did help us to improve the quality and clarity of this manuscript.